# Circulatory Shock among Hospitalized Patients for Salicylate Intoxication

**DOI:** 10.3390/diseases9010007

**Published:** 2021-01-12

**Authors:** Tananchai Petnak, Charat Thongprayoon, Wisit Kaewput, Fawad Qureshi, Boonphiphop Boonpheng, Saraschandra Vallabhajosyula, Tarun Bathini, Michael A. Mao, Ploypin Lertjitbanjong, Wisit Cheungpasitporn

**Affiliations:** 1Division of Pulmonary and Pulmonary Critical Care Medicine, Department of Medicine, Faculty of Medicine, Ramathibodi Hospital, Mahidol University, Bangkok 10400, Thailand; petnak@yahoo.com; 2Division of Pulmonary and Critical Care Medicine, Department of Medicine, Mayo Clinic, Rochester, MN 55905, USA; 3Division of Nephrology and Hypertension, Mayo Clinic, Rochester, MN 55905, USA; Qureshi.Fawad@mayo.edu; 4Department of Military and Community Medicine, Phramongkutklao College of Medicine, Bangkok 10400, Thailand; 5Department of Medicine, David Geffen School of Medicine, University of California, Los Angeles, CA 90095, USA; boonpipop.b@gmail.com; 6Section of Interventional Cardiology, Division of Cardiovascular Medicine, Department of Medicine, Emory University School of Medicine, Atlanta, GA 30322, USA; saraschandra21@gmail.com; 7Department of Internal Medicine, University of Arizona, Tucson, AZ 85724, USA; tarunjacobb@gmail.com; 8Division of Nephrology and Hypertension, Mayo Clinic, Jacksonville, FL 32224, USA; mao.michael@mayo.edu; 9Division of Pulmonary, Critical Care and Sleep Medicine, University of Tennessee Health Science Center, Memphis, TN 38163, USA; ploypinlert@gmail.com

**Keywords:** salicylate intoxication, salicylate, circulatory shock, epidemiology, outcomes, hospitalization

## Abstract

Background: This study aimed to evaluate the risk factors for circulatory shock and its impact on outcomes in patients hospitalized for salicylate intoxication. Methods: We used the National Inpatient Sample to identify patients hospitalized primarily for salicylate intoxication from 2003–2014. Circulatory shock was identified based on hospital diagnosis code for any type of shock or hypotension. We compared clinical characteristics, in-hospital treatments, outcomes, and resource use between patients with and without circulatory shock associated with salicylate intoxication. Results: Of 13,805 hospital admissions for salicylate intoxication, circulatory shock developed in 484 (4%) admissions. Risk factors for development of circulatory shock included older age, female sex, concurrent psychotropic medication overdose, anemia, congestive heart failure, volume depletion, rhabdomyolysis, seizure, gastrointestinal bleeding, and sepsis. Circulatory shock was significantly associated with increased odds of any organ failure and in-hospital mortality. Length of hospital stay and hospitalization cost was significantly higher in patients with circulatory shock. Conclusion: Approximately 4% of patients admitted for salicylate intoxication developed circulatory shock. Circulatory shock was associated with worse clinical outcomes and increased resource use.

## 1. Introduction

Salicylate has been widely available as an analgesic drug for several decades. Acetylsalicylate, also known as aspirin, is extensively utilized for its antiplatelet, antipyretic, and analgesic effects. Another derivative of salicylate is methyl salicylate, a major component of topical analgesic products [1]. As most salicylate-containing medications are over-the-counter products, salicylate intoxications have become one of the most common types of drug overdoses. In 2018, approximately 27,000 patients had overdoses of acetylsalicylate or methyl salicylate in the United States. The mortality rate of salicylate intoxication was 0.4% [2].

Salicylate intoxication can be categorized as either acute or chronic. Acute intoxication is usually seen in young adults after suicidal attempts, while chronic intoxication is more common in elderly patients with long-term aspirin use [3]. Clinical manifestations of acute salicylate intoxication vary by serum salicylate levels and acuity. Patients with serum salicylate levels ranging from 30 to 50 mg/dL may have mild symptoms such as hyperpnea, nausea, vomiting, or dizziness. In contrast, serum salicylate levels between 50 and 70 mg/dL are associated with more severe symptoms such as fevers, diaphoresis, tinnitus, and dehydration. Patients with serum salicylate levels exceeding more than 75 mg/dL, may present with coma, seizure, hallucination, tachypnea, cerebral edema, arrhythmias, heart failure, hypotension, renal failure, and coagulopathy [1]. However, clinical manifestations are not always dependent on the serum salicylate levels. The acuity of intoxication, patient comorbidities, and metabolism also plays a role. Patients with chronic salicylate intoxication usually manifest with nonspecific and more subtle neurologic symptoms [3], making the diagnosis of chronic salicylate intoxication challenging.

The primary pathogenesis of salicylate intoxication is the uncoupling of oxidative phosphorylation interfering with aerobic cellular respiration [3]. In addition, salicylates can affect the cardiovascular system. These cardiovascular effects may not be well recognized compared to other organ manifestations. A prior animal study showed that salicylate injection results in increased myocardial contraction, tachycardia, and vasodilatation [4]. In addition to the direct cardiovascular effects of salicylate, salicylate intoxication may induce hypovolemia through cutaneous fluid loss from sweating and gastrointestinal loss from emesis leading to hypovolemic circulatory shock [1].

We previously described the prevalence, clinical characteristics, outcomes, and resource use of hospitalization for salicylate intoxication in the United States [5]. We also assessed independent predictors for in-hospital mortality. Even though circulatory shock is associated with poor outcomes in several diseases [6,7], risk factors and impact of circulatory shock on outcomes and resource use in salicylate intoxication have not been well investigated. Therefore, we conducted this study to evaluate risk factors for circulatory shock and its impact on outcomes in hospitalized patients with salicylate intoxication in the United States.

## 2. Materials and Methods

### 2.1. Data Source

This cohort study utilized the National Inpatient Sample (NIS) database, the largest all-payer inpatient database in the United States. The Healthcare Cost and Utilization Project (HCUP), under the sponsorship of the Agency for Healthcare Research and Quality (AHRQ), manages and maintains the NIS database. The NIS database contains hospitalization data from a 20% stratified sample of United States hospitals. The patient-level information includes diagnostic and procedure codes. The institutional review board approval was waived as the data was from a de-identified public database.

### 2.2. Study Population

Hospitalized patients with a primary diagnosis of salicylate intoxication were identified using International Classification of Diseases, Ninth Revision, Clinical Modification (ICD-9 CM) diagnosis code 965.1. Hospitalized patients with salicylate intoxication were grouped based on the presence of circulatory shock. We identified circulatory shock as any type of shock or hypotension, using ICD-9 diagnosis codes of 785.5 (shock without mention of trauma), 785.50 (shock, unspecified), 785.51 (cardiogenic shock), 785.52 (septic shock), 785.59 (other shock without trauma, including hypovolemic shock), 458.8 (other specified hypotension), 458.9 (hypotension, unspecified), and 796.3 (nonspecific low blood pressure reading).

### 2.3. Data Collection

Clinical characteristics recorded included age, sex, race, year of hospitalization, alcohol use, concurrent analgesic overdose, psychotropic medication use, certain comorbidities (obesity, anemia, diabetes mellitus, hypertension, dyslipidemia, coronary artery disease, congestive heart failure, atrial fibrillation, chronic kidney disease), and certain acute medical conditions (volume depletion, rhabdomyolysis, seizure, gastrointestinal bleeding, sepsis, cardiac arrest). Recorded treatments included gastric lavage, invasive mechanical ventilation, blood transfusion, and dialysis. Outcomes consisted of types of organ failure (renal, respiratory, liver, neurological, and hematological) and in-hospital mortality. Resource use consisted of length of hospital stay and hospitalization cost. Clinical characteristics, treatments, and outcomes during hospitalization were identified using ICD-9 codes (Appendix A).

### 2.4. Statistical Analysis

Difference in clinical characteristics, treatments, outcomes, and resource use between hospitalized salicylate intoxication patients with and without circulatory shock was tested using Student’s *t*-test for continuous variables and Chi-squared test for categorical variables. Multivariable logistic regression with forward stepwise selection was performed to evaluate independent risk factors for circulatory shock. The associations of circulatory shock with treatments and outcomes were evaluated using logistic regression analysis. The associations of circulatory shock with resource use were evaluated using linear regression analysis. The analysis was adjusted for clinical characteristics that significantly differed between patients with and without circulatory shock (*p* < 0.05) in univariate analysis. Analysis was statistically significant when two-tailed *p*-value < 0.05. SPSS statistical software (version 22.0, IBM Corporation, Armonk, NY, USA) was used for all analyses.

## 3. Results

### 3.1. Incidence and Risk Factors for Circulatory Shock in Hospitalized Salicylate Intoxication Patients

Of 13,805 hospital admissions for salicylate intoxication, 484 (3.5%) developed circulatory shock during hospitalization. Table 1 compares clinical characteristics, in-hospital treatments, outcomes, and resource use between salicylate intoxication patients with and without circulatory shock.

Multivariable analysis identified older age, female sex, concurrent psychotropic medication overdose, anemia, congestive heart failure, volume depletion, rhabdomyolysis, seizure, gastrointestinal bleeding, and sepsis as independent risk factors for development of circulatory shock in salicylate intoxication patients (Table 2 and Figure 1).

### 3.2. The Association of Circulatory Shock with In-Hospital Treatments, Outcomes, and Resource Use

After adjusting for differences in baseline clinical characteristics, salicylate intoxication patients with circulatory shock required more invasive mechanical ventilation (OR 5.01; *p* < 0.001), blood transfusion (OR 3.12; *p* < 0.001), and dialysis (OR 3.04; *p* < 0.001). Circulatory shock in salicylate intoxication patients was significantly associated with increased odds of renal failure (OR 2.59; *p* < 0.001), respiratory failure (OR 4.39; *p* < 0.001), liver failure (OR 3.04; *p* < 0.001), neurological failure (OR 1.75; *p* < 0.001), hematological failure (OR 2.71; *p* < 0.001), and in-hospital mortality (OR 3.43; *p* < 0.001). In addition, circulatory shock was significantly associated with increased average length of hospital stay by 1.7 days and average hospitalization cost by USD 18,801 (Table 3).

## 4. Discussion

This is a large cohort study evaluating risk factors for circulatory shock and its impact among hospitalized patients with salicylate intoxication. Risk factors identified included sepsis, older age, recent year of admission, volume depletion, obesity, rhabdomyolysis, congestive heart failure, gastrointestinal bleeding, seizure, concurrent psychotropic agent overdose, and female sex. Patients with salicylate intoxication who developed circulatory shock had a higher risk of in-hospital mortality and associated organ failures. Circulatory shock was also associated with increased hospital resource utilization, including invasive mechanical ventilation, blood product transfusions, renal replacement therapy, length of hospital stay, and hospitalization cost.

The incidence of circulatory shock in hospitalized salicylate intoxication patients was approximately 3.6%. As noted above, there is currently limited data on circulatory shock in salicylate intoxication. Our study demonstrated that sepsis was the strongest predictor of circulatory shock in salicylate intoxication with adjusted odds ratio of 9.41 (95% CI 6.28–14.08). While there is no clear association between sepsis and salicylate intoxication, salicylate intoxication may similarly present with the systemic inflammatory response syndrome (SIRS) [8,9]. SIRS or sepsis may lead to vasodilatory shock through several mechanisms, including activation of ATP-sensitive potassium channels in vascular smooth muscles, enhancing nitric oxide synthase, and vasopressin deficiency [10]. As these pathophysiologic mechanisms for vasodilatory shock are shared between sepsis and salicylate intoxication, the further potential of hypovolemia with salicylate intoxication may further exacerbate the risk of circulatory shock when these conditions coexist. Hypovolemia can develop in salicylate intoxication due to diaphoresis and emesis, with up to 4–6 L of volume deficit in severe intoxications [1,3]. Similarly, the association of increased circulatory shock in salicylate intoxication patients with volume depletion, rhabdomyolysis, and gastrointestinal bleeding may occur due to similar cumulative insults to the cardiovascular system and effective circulating volume. Overdoses of psychotropic medications may increase risk of circulatory shock in salicylate intoxication patients via the vasodilatative effects of α_1_-adrenergic receptor blockage [11].

Our study demonstrated that older age also increased the risk of circulatory shock. Diagnosis of salicylate intoxication in elderly patients can be challenging. Elderly patients more often present with chronic salicylate intoxication from long-term aspirin use. Chronic intoxication frequently manifests with atypical and nonspecific symptoms such as altered mental status and delirium [1]. Delayed diagnosis and treatment of these patients can increase the severity of the disease and its complications [12]. Therefore, increased awareness of salicylate intoxication in general public and health care providers is the key for prevention as well as prompt diagnosis and treatment of salicylate intoxication. Obesity was also associated with a higher risk of circulatory shock in salicylate intoxication. Salicylate intoxication can increase body temperature by uncoupling of oxidative phosphorylation in the mitochondria. It can be hypothesized that obese patients tend to have higher heat production and fluid loss from diaphoresis, translating to increased risk for circulatory shock [6]. In addition, elderly and obese patients have been observed to have lower salicylate clearances [13]. Therefore, the severity of salicylate intoxication and its complications may be enhanced in these patients. Other risk factors for circulatory shock included seizure, congestive heart failure, and female sex.

Circulatory shock has been associated with increased mortality in several medication conditions [6,7]. Data has been limited, however, on its impact on hospital outcomes and resource utilization in patients with salicylate intoxication. Our study showed that circulatory shock was associated with higher in-hospital mortality with adjusted odds ratio of 3.43 (95% CI 1.88–6.24). Moreover, salicylate intoxication patients with circulatory shock had increased risk of other organ failures, including respiratory, liver, hematologic, renal, and neurologic, with adjusted odds ratios of 4.39, 3.04, 2.71, 2.59, and 1.75, respectively. Salicylate intoxication patients may develop tachypnea and respiratory failure due to central nervous system direct respiratory center stimulation and compensatory chemoreceptor response from metabolic acidosis. Salicylate intoxication has been previously independently associated with other organ system complications such as renal failure, liver injury, coagulopathy, and neurologic alterations [1,3]. Concurrent circulatory failure may either increase risk for other organ failures directly or it may serve as an indicator for the severity of salicylate intoxication and its detrimental effects.

Circulatory shock in salicylate intoxication increased the need for invasive medical interventions, including mechanical ventilation and renal replacement therapy. As described above, circulatory shock increased the risk of respiratory and renal failure. Therefore, circulatory shock should be associated with a higher risk of invasive mechanical ventilation and renal replacement therapy. Finally, we demonstrated that circulatory shock was associated with higher resource utilization, longer length of stay, and increased hospitalization costs compared to salicylate intoxication in patients without circulatory shock. Although the impact of circulatory shock on hospital resource utilization has never been previously described in salicylate intoxication patients, circulatory shock has been previously reported to increase hospital resource utilization in other conditions, such as heat stroke [6].

Even though our study comprised a large cohort of salicylate intoxication patients, some limitations should be noted. First, the NIS database is a database limited to hospitalizations. Accordingly, our study could not define long-term outcomes of salicylate intoxication patients with circulatory shock. In addition, the incidence of circulatory shock in salicylate intoxication may be an overestimation, as patients with milder severity of intoxication may not have been admitted to the hospital. Second, we could not conclude the direction of the association between circulatory shock and other complications due to the lack of time sequence in the NIS database. Third, the diagnosis of circulatory shock is based on diagnosis codes. The diagnosis of circulatory shock and the codes themselves may represent heterogeneous disease, as there were no definitive diagnostic criteria utilized. Fourth, although it is known that majority of hospitalized patients with salicylate toxicity had acute intoxication [5,14], we could not categorize salicylate intoxication into acute or chronic intoxication, which may lead to different patient characteristics and outcome results. However, patients with chronic salicylate intoxication usually present with nonspecific, delirium, confusion, dehydration, fever, and unexplained metabolic acidosis patients, rather than cerebral edema, pulmonary edema, and circulatory shock that are more common in patients with acute salicylate intoxication. Lastly, the dosage and duration of, and reason for aspirin use could not be determined in the NIS database. Therefore, we could not assess the relationship of the dosage or duration of aspirin use with circulatory shock.

## 5. Conclusions

In summary, circulatory shock is not a common complication of salicylate intoxication. Circulatory shock was associated with an increased risk of in-hospital mortality and other organ failures. The risk factors for circulatory shock in salicylate intoxication included sepsis, older age, recent year of admission, volume depletion, obesity, rhabdomyolysis, congestive heart failure, gastrointestinal bleeding, seizure, combined psychotropic agent overdose, and female sex.

## Figures and Tables

**Figure 1 diseases-09-00007-f001:**
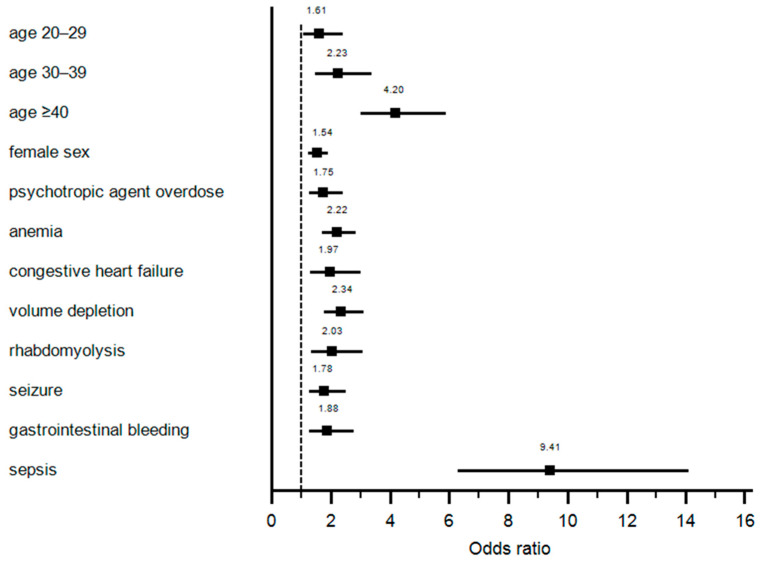
Independent risks factors for development of circulatory shock.

**Table 1 diseases-09-00007-t001:** Clinical characteristics, in-hospital treatments, outcomes, and resource utilization in salicylate intoxication patients with and without circulatory shock.

	Total	Circulatory Shock	No Circulatory Shock	*p*-Value
**Clinical Characteristics**				
N (%)	13,805	484	13,321	
Age (years), mean ± SD	34.0 ± 18.7	47.6 ± 19.0	33.5 ± 18.5	<0.001
<20	3902 (28.3)	42 (8.7)	3860 (29.0)	<0.001
20–29	3228 (23.4)	61 (12.6)	3167 (23.8)	
30–39	1951 (14.1)	56 (11.6)	1895 (14.2)	
≥40	4710 (34.2)	325 (67.1)	4385 (33.0)	
Female	8994 (65.0)	344 (71.1)	8650 (64.9)	0.005
Race				<0.001
Caucasian	7729 (56.0)	319 (65.9)	7410 (55.6)	
African American	1391 (10.1)	39 (8.0)	1352 (10.1)	
Hispanic	1311 (9.5)	38 (7.9)	1273 (9.6)	
Asian or Pacific Islander	200 (1.4)	4 (0.8)	196 (1.5)	
Other	3174 (23.0)	84 (17.4)	3090 (23.2)	
Year of hospitalization				<0.001
2003–2006	5011 (36.3)	86 (17.8)	4925 (37.0)	
2007–2010	4434 (32.1)	156 (32.2)	4278 (32.1)	
2011–2014	4360 (31.6)	242 (50.0)	4118 (30.9)	
Alcohol drinking	2216 (16.1)	73 (15.1)	2143 (16.1)	0.55
Analgesics overdose	967 (7.0)	37 (7.6)	930 (7.0)	0.57
Psychotropic agent overdose	896 (6.5)	48 (9.9)	848 (6.4)	0.002
Obesity	521 (3.8)	15 (3.1)	506 (3.8)	0.43
Anemia	897 (6.5)	98 (20.2)	799 (6.0)	<0.001
Diabetes Mellitus	801 (5.8)	49 (10.1)	752 (5.6)	<0.001
Hypertension	2137 (15.5)	129 (26.7)	2008 (15.1)	<0.001
Dyslipidemia	749 (5.4)	35 (7.2)	714 (5.4)	0.07
Coronary artery disease	512 (3.7)	60 (12.4)	452 (3.4)	<0.001
Congestive heart failure	239 (1.7)	31 (6.4)	208 (1.6)	<0.001
Atrial flutter/fibrillation	172 (1.2)	20 (4.1)	152 (1.1)	<0.001
Chronic kidney disease	218 (1.6)	27 (5.6)	191 (1.4)	<0.001
Liver cirrhosis	116 (0.8)	5 (1.0)	111 (0.8)	0.61
Volume depletion	739 (5.4)	67 (13.8)	672 (5.0)	<0.001
Rhabdomyolysis	258 (1.9)	33 (6.8)	225 (1.7)	<0.001
Seizure	565 (4.1)	45 (9.3)	520 (3.9)	<0.001
Gastrointestinal bleeding	363 (2.6)	38 (7.9)	325 (2.4)	<0.001
Sepsis	126 (0.9)	48 (9.9)	78 (0.6)	<0.001
Cardiac arrest	95 (0.7)	14 (2.9)	81 (0.6)	<0.001
**Treatments**				
Gastric lavage	344 (2.5)	18 (3.7)	326 (2.4)	0.08
Noninvasive ventilation	64 (0.5)	7 (1.4)	57 (0.4)	0.007
Invasive mechanical ventilation	760 (5.5)	153 (31.6)	607 (4.6)	<0.001
Blood component transfusion	356 (2.6)	68 (14.0)	288 (2.2)	<0.001
Renal replacement therapy	811 (5.9)	116 (24.0)	695 (5.2)	<0.001
**Outcomes**				
Renal failure	1279 (9.3)	157 (32.4)	1122 (8.4)	<0.001
Respiratory failure	943 (6.8)	168 (34.7)	775 (5.8)	<0.001
Liver failure	110 (0.8)	20 (4.1)	90 (0.7)	<0.001
Neurological failure	689 (5.0)	64 (13.2)	625 (4.7)	<0.001
Hematological failure	303 (2.2)	48 (9.9)	255 (1.9)	<0.001
In-hospital mortality	132 (1.0)	26 (5.4)	106 (0.8)	<0.001
**Resource Utilization**				
Length of hospital stay (days), mean ± SD	2.6 ± 3.3	6.0 ± 7.3	2.4 ± 3.0	<0.001
Hospitalization cost (USD), mean ± SD	18,128 ± 29,613	52,933 ± 83,537	16,856 ± 24,668	<0.001

**Table 2 diseases-09-00007-t002:** Factors associated with circulatory shock in salicylate intoxication patients.

Variables	Univariable Analysis		Multivariable Analysis	
	Crude Odds Ratio (95%CI)	*p*-Value	Adjusted Odds Ratio (95%CI)	*p*-Value
Age (years)				
<20	1 (reference)		1 (reference)	
20–29	1.77 (1.19–2.63)	0.005	1.61 (1.08–2.40)	0.02
30–39	2.72 (1.81–4.07)	<0.001	2.23 (1.48–3.37)	<0.001
≥40	6.81 (4.93–9.42)	<0.001	4.20 (3.00–5.88)	<0.001
Female	1.33 (1.09–1.64)	0.005	1.54 (1.25–1.92)	<0.001
Race				
Caucasian	1 (reference)			
African American	0.67 (0.48–0.94)	0.02		
Hispanic	0.69 (0.49–0.98)	0.04		
Asian or Pacific Islander	0.47 (0.18–1.28)	0.14		
Other	0.63 (0.50–0.81)	<0.001		
Year of data collection				
2003–2006	1 (reference)		1 (reference)	
2007–2010	2.09 (1.60–2.73)	<0.001	1.88 (1.43–2.47)	<0.001
2011–2014	3.37 (2.62–4.32)	<0.001	2.86 (2.21–3.70)	<0.001
Alcoholic drinking	0.93 (0.72–1.19)	0.55		
Analgesics overdose	1.10 (0.78–1.55)	0.58		
Psychotropic agent overdose	1.62 (1.19–2.20)	0.002	1.75 (1.27–2.40)	0.001
Obesity	0.81 (0.48–1.37)	0.43		
Anemia	3.98 (3.15–5.02)	<0.001	2.22 (1.72–2.86)	<0.001
Diabetes Mellitus	1.88 (1.39–2.55)	<0.001		
Hypertension	2.05 (1.66–2.52)	<0.001		
Dyslipidemia	1.38 (0.97–1.96)	0.08		
Coronary artery disease	4.03 (3.03–5.36)	<0.001		
Congestive heart failure	4.31 (2.93–6.36)	<0.001	1.97 (1.29–3.01)	0.002
Atrial flutter/fibrillation	3.73 (2.32–6.01)	<0.001		
Chronic kidney disease	4.06 (2.69–6.14)	<0.001		
Liver cirrhosis	1.24 (0.51–3.06)	0.64		
Volume depletion	3.02 (2.31–3.96)	<0.001	2.34 (1.76–3.11)	<0.001
Rhabdomyolysis	4.26 (2.92–6.21)	<0.001	2.03 (1.34–3.07)	0.001
Seizure	2.52 (1.83–3.47)	<0.001	1.78 (1.27–2.50)	0.001
Gastrointestinal bleeding	3.41 (2.40–4.83)	<0.001	1.88 (1.27–2.77)	0.001
Sepsis	18.69 (12.89–27.12)	<0.001	9.41 (6.28–14.08)	<0.001
Cardiac arrest	4.87 (2.74–8.65)	<0.001		
Gastric lavage	1.54 (0.95–2.50)	0.08		

**Table 3 diseases-09-00007-t003:** The association of circulatory shock with in-hospital treatments, complications, outcomes, and resource utilization in salicylate intoxication patients.

Variables	Univariable Analysis		Multivariable Analysis	
	Crude Odds Ratio (95%CI)	*p*-Value	Adjusted Odds Ratio * (95%CI)	*p*-Value
**Treatment**				
Noninvasive ventilation	3.42 (1.55–7.53)	0.002	1.01 (0.41–2.49)	0.98
Invasive mechanical ventilation	9.68 (7.86–11.92)	<0.001	5.01 (3.89–6.45)	<0.001
Blood component transfusion	7.40 (5.58–9.80)	<0.001	3.12 (2.19–4.44)	<0.001
Renal replacement therapy	5.73 (4.59–7.15)	<0.001	3.04 (2.37–3.90)	<0.001
**Complication and Outcome**				
Renal failure	5.22 (4.27–6.38)	<0.001	2.59 (2.04–3.28)	<0.001
Respiratory failure	8.61 (7.04–10.52)	<0.001	4.39 (3.45–5.59)	<0.001
Liver failure	6.34 (3.87–10.38)	<0.001	3.04 (1.73–5.35)	<0.001
Neurological failure	3.10 (2.35–4.08)	<0.001	1.75 (1.30–2.36)	<0.001
Hematological failure	5.64 (4.09–7.79)	<0.001	2.71 (1.89–3.88)	<0.001
In-hospital mortality	7.08 (4.56–10.98)	<0.001	3.43 (1.88–6.24)	<0.001
**Resource Utilization**	**Coefficient** **(95% CI)**	***p*-Value**	**Adjusted Coefficient** **(95% CI)**	***p*-Value**
Length of hospital stay (days)	3.5 (3.2–3.8)	<0.001	1.7 (1.4–2.0)	<0.001
Hospitalization cost (USD)	36,078 (33,449–38,706)	<0.001	18,801 (16,346–21,255)	<0.001

Notes: * adjusted for age, sex, race, year, psychotropic agent overdose, anemia, diabetes mellitus, hypertension, coronary artery disease, congestive heart failure, atrial fibrillation, chronic kidney disease, rhabdomyolysis, seizure, volume depletion disorder, gastrointestinal bleeding, sepsis, and ventricular arrhythmia/cardiac arrest.

## Data Availability

Restrictions apply to the availability of these data. Data was obtained from the Healthcare Cost and Utilization Project (HCUP) under the sponsorship of the Agency for Healthcare Research and Quality (AHRQ), and are available https://www.distributor.hcup-us.ahrq.gov/ with the permission of HCUP/AHRQ.

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
