# Peer review of "Circulatory Shock among Hospitalized Patients for Salicylate Intoxication"

_diseases, 2021, doi:10.3390/diseases9010007_

Round 1
Reviewer 1 Report
In the manuscript by Petnak et al., the authors analyzed the risk factors for the circulatory shock in patients hospitalized for salicylate intoxication.
Generally, the structure of the proposed manuscript is presented clearly and readably.
The findings are interesting but the authors have already published a paper (Hospitalizations for Acute Salicylate Intoxication in the United States. Thongprayoon C, Petnak T, Kaewput W, Mao MA, Kovvuru K, Kanduri SR, Boonpheng B, Bathini T, Vallabhajosyula S, Pivovarova AI, Brar HS, Medaura J, Cheungpasitporn W. J Clin Med. 2020 Aug 14;9(8):2638) where they analyzed almost the same parameters. The novelty of the presented investigations is not enough stressed.
The study must be improved by better structuring of the material to give significant contribution to the field. As it is, I cannot recommend acceptance of this manuscript for publication.
Author Response
Response to reviewer #1
In the manuscript by Petnak et al., the authors analyzed the risk factors for the circulatory shock in patients hospitalized for salicylate intoxication. Generally, the structure of the proposed manuscript is presented clearly and readably.
The findings are interesting but the authors have already published a paper (Hospitalizations for Acute Salicylate Intoxication in the United States. Thongprayoon C, Petnak T, Kaewput W, Mao MA, Kovvuru K, Kanduri SR, Boonpheng B, Bathini T, Vallabhajosyula S, Pivovarova AI, Brar HS, Medaura J, Cheungpasitporn W. J Clin Med. 2020 Aug 14;9(8):2638) where they analyzed almost the same parameters. The novelty of the presented investigations is not enough stressed.
The study must be improved by better structuring of the material to give significant contribution to the field. As it is, I cannot recommend acceptance of this manuscript for publication.
Response: We acknowledged that our current study used the same cohort as our previously published study and there were overlapping variables in these two studies. However, our current study has distinct objectives and findings from previously published study, as shown below, which contributed additional knowledge to the field and worthy separate publication. We did not initially cite previous publication in our current manuscript because it had not been published at the time of our manuscript submission.
The objectives and findings of previous publication were
- Describe the prevalence and trend of hospitalization for salicylate intoxication
Finding the overall inpatient prevalence of salicylate intoxication during study period of 2003-2014 was 148 cases per 1,000,000 admissions, ranging from 128 to 167 cases per 1,000,000
- Describe clinical characteristics, outcomes, resource use of hospitalization for salicylate intoxication
Finding The mean age was 34 years. 35% were male. 65% used salicylate for suicidal attempts. 6% required renal replacement therapy. The most common complication of salicylate intoxication was electrolyte and acid-base disorders, while the most common organ dysfunction was kidney failure. In-hospital mortality was 1.0%. The mean length of hospital stay was 2 days. The median hospitalization cost was $11,172.
- Assess independent predictors for in-hospital mortality in hospitalization for salicylate intoxication
Finding Independent predictors for increased in-hospital mortality was older age, Asian/Pacific Islander race, diabetes mellitus, hyponatremia, ventricular arrhythmia, kidney failure, respiratory failure, and neurological failure, while predictors for decreased in-hospital mortality was African American and Hispanic race.
The objectives and findings of current manuscript were
- Describe the incidence of circulatory shock in hospitalization for salicylate intoxication
Finding The incidence of circulatory shock was 4% in hospitalized salicylate intoxication patients.
- Assess risk factors for development of circulatory shock in hospitalization for salicylate intoxication.
Finding Risk factors for development of circulatory shock in hospitalized salicylate intoxication patients included older age, female sex, concurrent psychotropic medication overdose, anemia, congestive heart failure, volume depletion, rhabdomyolysis, seizure, gastrointestinal bleeding, and sepsis
- Evaluate the impact of circulatory shock on outcomes, and resource use in hospitalization for salicylate intoxication.
Finding Circulatory shock was significantly associated with increased organ dysfunction and mortality, longer hospital length of stay, and higher hospitalization cost.
The following statements have been added to introduction to highlight the novelty of present investigation.
“We previously described the prevalence, clinical characteristics, outcomes, and resource use of hospitalization for salicylate intoxication in United States [5]. We also assessed independent predictors for in-hospital mortality. Even though circulatory shock is associated with poor outcomes in several diseases [6,7], risk factors and impact of circulatory shock on outcomes and resource use in salicylate intoxication have not been well investigated.”
We greatly appreciated the reviewer’s and editor’s time and comments to improve our manuscript. The manuscript has been improved considerably by the suggested revisions.

Reviewer 2 Report
This is an interesting paper analysing available data from a sizeable cohort of patients. The paper kept me engaged, and the layout was clear.
Of interest is the analysis of chronic aspirin use in the elderly.
Is it possible to tease out how long these patients had been on aspirin?
hiw long is required for chronic usage? Is this from dosage given for post-CVD diagnosis?
Author Response
Response to reviewer #2
This is an interesting paper analyzing available data from a sizeable cohort of patients. The paper kept me engaged, and the layout was clear. Of interest is the analysis of chronic aspirin use in the elderly.
Response: We thank the reviewer for your helpful comments.
Comment #1
Is it possible to tease out how long these patients had been on aspirin?
How long is required for chronic usage? Is this from dosage given for post-CVD diagnosis?
Response: The reviewer raises important point. Data on the dosage and duration of, and reason for aspirin use are limited and thus could not be determined in the NIS database The following statements have been added to the limitation section.
“Lastly, the dosage and duration of, and reason for aspirin use could not be determined in the NIS database. Therefore, we could not assess the relationship of the dosage or duration of aspirin use with circulatory shock.”
We greatly appreciated the reviewer’s and editor’s time and comments to improve our manuscript. The manuscript has been improved considerably by the suggested revisions.

Round 2
Reviewer 1 Report
The manuscript by Petnak et al., has certainly been improved but I believe that the reader would benefit from a summary diagram, where all of the findings are brought together. After these modifications, I can recommend the acceptance of this manuscript for publication
Author Response
Response to Reviewer 1
The manuscript by Petnak et al., has certainly been improved but I believe that the reader would benefit from a summary diagram, where all of the findings are brought together. After these modifications, I can recommend the acceptance of this manuscript for publication
Response: We appreciate the reviewer’s input. We agree with the review. We added Figure 1 to summarize risk factors for development of circulatory shock, as suggested (attached PDF) and in the revised manuscript.
